# Antioxidative and Anti-Inflammatory Phytochemicals and Related Stable Paramagnetic Species in Different Parts of Dragon Fruit

**DOI:** 10.3390/molecules26123565

**Published:** 2021-06-10

**Authors:** Chalermpong Saenjum, Thanawat Pattananandecha, Kouichi Nakagawa

**Affiliations:** 1Cluster of Excellence on Biodiversity-Based Economic and Society (B.BES-CMU), Chiang Mai University, Chiang Mai 50200, Thailand; thanawat.pdecha@gmail.com; 2Department of Pharmaceutical Sciences, Faculty of Pharmacy, Chiang Mai University, Chiang Mai 50200, Thailand; 3Division of Regional Innovation, Graduate School of Health Sciences, Hirosaki University, 66-1 Hon-Cho, Hirosaki 036-8564, Japan

**Keywords:** EPR, HPLC, anthocyanins, dragon fruits, antioxidant, anti-inflammatory

## Abstract

In this study, we investigated the antioxidant and anti-inflammatory phytochemicals and paramagnetic species in dragon fruit using high-performance liquid chromatography (HPLC) and electron paramagnetic resonance (EPR). HPLC analysis demonstrated that dragon fruit is enriched with bioactive phytochemicals, with significant variations between each part of the fruit. Anthocyanins namely, cyanidin 3-glucoside, delphinidin 3-glucoside, and pelargonidin 3-glucoside were detected in the dragon fruit peel and fresh red pulp. Epicatechin gallate, epigallocatechin, caffeine, and gallic acid were found in the dragon fruit seed. Additionally, 25–100 mg × L^−1^ of dragon fruit pulp and peel extracts containing enrichment of cyanidin 3-glucoside were found to inhibit the production of reactive oxygen species (ROS), reactive nitrogen species (RNS), inducible nitric oxide synthase (iNOS), and cyclooxygenase-2 (COX-2) in cell-based studies without exerted cytotoxicity. EPR primarily detected two paramagnetic species in the red samples. These two different radical species were assigned as stable radicals and Mn^2+^ (paramagnetic species) based on the g-values and hyperfine components. In addition, the broad EPR line width of the white peel can be correlated to a unique moiety in dragon fruit. Our EPR and HPLC results provide new insight regarding the phytochemicals and related stable intermediates found in various parts of dragon fruit. Thus, we suggest here that there is the potential to use dragon fruit peel, which contains anthocyanins, as a natural active pharmaceutical ingredient.

## 1. Introduction

Dragon fruit (*Hylocereus* sp.), commonly known as pitaya or pitahaya, is a member of the cactus family Cactaceae [1]. It is native to the tropical forest region of Mexico and Central South America [2]. The peel of the fruit is usually red, with white (*Hylocereus undatus*) and red (*H. polyrhizus*, *H. costaricensis*) flesh. The fruit is a rich source of nutrients and minerals, including vitamins B_1_, B_2_, B_3_, and C, protein, fat, carbohydrate, crude fiber, thiamin, trigonelline, niacin, alanine, arginine, leucine/isoleucine, glutamate, glutamine, shikimate, rutin, gamma-aminobutyric acid (GABA), valine, choline, pyridoxine, kobalamin, glucose, sucrose, fructose, inositol, betacyanins, phosphorus, iron, formic acid, malic acid, citric acid, fumaric acid, succinic acid, azelaic acid, ascorbic acid, aspartic acid, betalamic acid, betanin, oleic acid, fatty acids, catechins, quercetin derivatives, caffeic acid, 2’-*O*-glucosyl-betanin, hylocerenin, 2’-*O*-apiosyl-betanin, phytoalbumin, 2’-*O*-apiosyl-phyllocactin, and 2’-(5’-*O*-E-feruloylapiosyl) betanin [3,4,5,6]. Fiber and vitamins promote the proper functioning of the gastrointestinal system, prevent colorectal cancer and diabetes, remove or increase the secretion of toxic substances such as heavy metals, and control cholesterol levels and blood pressure [7,8]. Betacyanins and betaxanthins, water-soluble pigments in the peel and pulp of dragon fruit [9,10], are used as natural food colorants in the food industry [11]. The seed of dragon fruit also contains catechin, epicatechin, rutin, quercetin, and myricetin [12]. Anthocyanins can be found in the skin of *H. undatus*, including cyanidin 3-*O*-glucoside, cyanidin 3,5-*O*-glucoside, and pelargonidin 3,5-*O*-glucoside [13]. However, much remains unknown about the paramagnetic species (Mn^2+^, antioxidative organic radicals, etc.) present in dragon fruits and their antioxidant and anti-inflammatory activities.

Electron paramagnetic resonance (EPR) spectroscopy is a very sensitive technique that can non-destructively detect free radicals [14]. The magnetic-field position and relative intensities of the EPR lines occur between the energy levels of electrons with unpaired spins. Transitions occur between energy levels, which give rise to lines in the spectrum. The EPR spectrum appears either as a series of multiple overlapped lines or as an asymmetric line shape, depending on the paramagnetic species within the sample [15]. Numerous plants contain various antioxidants to reduce the damage caused by reactive oxygen species (ROS) and reactive nitrogen species (RNS). The antioxidative processes result in the production of antioxidant intermediates. Various antioxidants contain OH groups and conjugated double bonds. The delocalization of unpaired spin forms stable intermediates which are detected by EPR. Additionally, high-performance liquid chromatography (HPLC) is used to separate, identify, and quantitate various antioxidants in plants. However, there are no reports describing the distribution of antioxidative phytochemicals and related intermediates in dragon fruits.

In this investigation, the paramagnetic species in each part of the dragon fruit were examined using EPR and antioxidative phytochemicals were analyzed HPLC. Additionally, the antioxidative and anti-inflammatory activities were also investigated in a cell-based study. The EPR technique measured two types of paramagnetic species in dragon fruits, which displayed antioxidative effects. The EPR line widths and phytochemical activities of various parts of the dragon fruit were discussed.

## 2. Results and Discussion

### 2.1. TPC, TFC, and TAC Content

Polyphenolic compounds have been reported to be commonly found in both edible and inedible plants [16]. Flavonoid compounds are usually found in peels, seeds, and stems, while non-flavonoid compounds are located in pulps [17]. All parts of the dragon fruits have color, except the white pulp (Figure 1). As listed in Table 1, phenolic and flavonoid compounds were detected in all dragon fruit samples. Both red and white pulp seed were found to have significantly greater phenolic and flavonoid contents than the other parts (*p* < 0.05), which is similar to the findings of the study of Nguyen et al. [18], who reported on the total phenolic compounds in the seed and pulp of red and white dragon fruits. Moreover, Nurliyana et al. [19] also reported that the peels of both red dragon fruit (*H. polyrhizus*) and white dragon fruit (*H. undatus*) contained higher phenolic compounds than the pulps. The flavonoid content in red and white dragon fruits was reported to be higher in the peel when compared to the pulp [20]. Interestingly, the extracts from fresh white pulp showed very low levels of antioxidant phytochemicals. In addition, anthocyanin was found in fresh red pulp, red peel, and white peel. Anthocyanins are plant pigments that are responsible for the color of the pulp and the peel of dragon fruits. Malvidin, cyanidin, and delphinidin are anthocyanin compounds found in the peel of the red dragon fruit (*H. polyrhizus*) using liquid chromatography-mass spectrometry (LC-MS) assay [21].

Three sample extracts, including fresh purple pulp, red peel, and white peel were selected for the analysis of anthocyanin by HPLC. Red pulp seed and white pulp seed were also analyzed for catechin and related compounds by HPLC.

### 2.2. Chromatographic Analysis of Antioxidative Phytochemicals

The HPLC chromatogram identifying anthocyanins in RP compared with mixed anthocyanin standards, including delphinidin 3-glucoside, cyanidin 3-glucoside, peonidin 3-glucoside, pelargonidin 3-glucoside, delphinidin, and cyanidin, is shown in Figure 2. HPLC chromatograms indicated that cyanidin 3-glucoside, delphinidin 3-glucoside, and pelargonidin 3-glucoside in average concentrations of 12.67 ± 0.63, 0.82 ± 0.17, and 1.76 ± 0.23 mg/ 100 g dried samples, respectively, were found in the reddish parts of the dragon fruit, including fresh red pulp, red peel, and white peel. The results corresponded to those of Fan et al. [22], who reported that cyanidin 3-glucoside, cyanidin 3-rutinoside, and delphinidin 3-*O*-beta-d-glucoside were detected in the reddish coloration of dragon fruit. Moreover, cyanidin 3-*O*-glucoside, cyanidin 3,5-*O*-glucoside, and pelargonidin 3,5-*O*-glucoside were also identified in the peel of dragon fruit (*H. undatus*) extracted using 1% trifluoroacetic acid in methanol as an extraction solvent, with an average concentration of 44.3865 ± 1.3125 mg/100 g of the sample [13]. It has been suggested that extraction methods and physical, chemical, and biological processes affect the content of anthocyanins in dragon fruit parts due to enzymatic and non-enzymatic changes [21,23].

Figure 3 shows the HPLC chromatogram of RP-S compared with the mixed catechin and the related compounds. Among the 10 standards used in this HPLC system, gallic acid, epigallocatechin, catechin, caffeine, catechin, epicatechin, and epicatechin gallate were detected in the seeds of dragon fruit. Catechin and epicatechin were detected as the major components of the dragon fruit seeds. These results corresponded with those of Adnan et al. [12], who reported that catechin, quercetin, epicatechin, myricetin, and epicatechin were detected in red dragon fruit seed, while catechin was the major flavonoid detected, at 3.60 ± 2.33 mg/g dry weight followed by quercetin at 1.31 ± 0.45 mg/g dry weight. A recent study using ultra-performance liquid chromatography coupled with electrospray ionization quadrupole time-of-flight tandem mass spectrometry (UPLC-QTOF-MS/MS) analysis by Nguyen et al. [18] reported that isorhamnetin glycoside, rutin, quercetin hexoside, kaempferol glucorhamnoside, isorhamnetin, and galloylglucoside were detected in both red and white dragon fruit peels, while kaempferol glucorhamnoside, kaempferol glucoside, kaempferol 3-glucoside, gallic acid, and ellagic acid were detected in both red and white dragon fruit seeds, but only isorhamnetin glycoside and isorhamnetin were detected in red dragon fruit seed. There were differences in the content of antioxidative phytochemicals that may be affected by several factors, including fruit species, maturity stage of fruits, geographical location, growing and environmental conditions, and methods of cultivation, which can cause variations in the quality of bioactive compounds in fruits [24,25].

### 2.3. Determination of the Inhibitory Effects on Intracellular ROS and RNS Production

The antioxidant and free radical scavenging activities were strongly correlated with the content of total phenolic and total flavonoid compounds because of the mechanism of the hydrogen donation and electron transfer of both phenolic and flavonoid compounds to free radical molecules [26,27]. The inhibitory effects of the extracts of each part of the dragon fruit on nitric oxide production in LPS/IFN-γ-induced RAW 264.7 cells and on intracellular ROS production in PBMC cells are illustrated in Table 2. The fresh red pulp extract exhibited the highest inhibition of induced RNS and ROS production in cell-based studies. Both tested samples containing anthocyanin and catechin and related compounds exhibited inhibitory effects on intracellular ROS and RNS production. Interestingly, the tested samples containing anthocyanin including fresh red pulp, red peel, and white peel demonstrated a higher inhibition activity for induced RNS and ROS production compared with samples containing catechin and related compounds (red pulp seed and white pulp seed). These results corresponded with the report of Wang et al. [28] and Kuskoski et al. [29], who reported that anthocyanins namely, delphinidine and cyanidin 3-glucoside have higher antioxidative properties in comparison with known antioxidants such as catechin and vitamin E derivatives with the trolox equivalence antioxidant capacity (TEAC) 2.45 ± 0.45 and 1.85 ± 0.26 mM, respectively compared to the TEAC of peonidin 3-glucoside, pelargonidin 3-glucoside, and malvidin 3-glucoside in the value of 1.49 ± 0.19, 1.50 ± 0.08, and 1.41 ± 0.07 mM, respectively. The antioxidative property of anthocyanins depends on the number of free hydroxyl groups around the pyrone ring, while the position of the hydroxyl groups and the conjugated double bonds also plays an important role in antioxidant activity. Additionally, 3’,4’ di-OH group in B ring of delphinidine and cyanidine plays a key role in antioxidant and free radical scavenging activities [30].

### 2.4. Determination of Anti-Inflammatory Activities

Inducible nitric oxide synthase (iNOS) and cyclooxygenase-2 (COX-2) play a pivotal role in catalyzing the rate-limiting step in prostaglandin biosynthesis and in the mediation of the inflammation process. The inhibitory effects of the selected anthocyanin extracts, including fresh red pulp, red peel, and white peel, on iNOS and COX-2 production in LPS/IFN-γ-induced RAW 264.7 cells are illustrated in Figure 4 and Figure 5, respectively. The result revealed that the tested samples in the concentrations of 25 to 100 mg × L^−1^ exhibited an inhibitory effect on both iNOS and COX-2 production in LPS/IFN-γ-induced RAW 264.7 cells in a dose-dependent manner without any cytotoxicity. Our results correspond to those of Jung et al. [31], who reported that cyanidin-3-glucoside blackberry reduced nitric oxide and prostaglandin E2 production in LPS-stimulated RAW264.7 cells by values of 39.7% and 52.6%, respectively. Additionally, the protein expressions of iNOS and COX-2 in LPS-stimulated RAW264.7 cells were also decreased in cells treated with cyanidin-3-glucoside prepared from blackberry via down-regulated nuclear factor-kappa B (NF-κB) expression and up-regulated I-κB expression in LPS-stimulated macrophages. Moreover, cyanidin-3-glucoside enriched fraction prepared from black rice containing the total anthocyanin and total flavonoid content in the values of 8.1 ± 1.9 mg cyanidin 3-glucoside/g extract and 42.9 ± 2.1 mg catechin/g extract, respectively, also significantly inhibited the LPS-induced production of NO and the expression of iNOS and COX-2 in RAW 264.7 cells via the downregulation of the NF-kB and activator protein 1 (AP-1) signaling pathways [32]. Furthermore, cyanidin 3-glucoside prevented the tumor necrosis factor-αTNF-α) induced inflammation process in Caco-2 cells through increasing the translocation of the transcription factor nuclear factor erythroid 2-related factor 2 (Nrf2), a transcription factor that may regulate the expression of antioxidant proteins which protect against oxidative damage-induced by inflammatory cytokines, into the nucleus, thus activating antioxidant and detoxifying genes [33].

The overall results indicate that the cyanidin 3-glucoside-enriched extract prepared from the red pulp and peel of dragon fruit exhibited potent antioxidant effects and may suppress inflammatory response through the inhibition of NO and ROS production, and it has been proposed to modulate gene expression.

### 2.5. EPR of Dragon Fruits

The EPR spectrum of dragon fruit peel, pulp, and one to two whole seeds is shown in Figure 6. The EPR spectrum here is composed of two kinds of signals; these signals were relatively stable, with no depreciation in the signal for at least a month when the samples were kept at 5 °C. One signal corresponds to Mn^2+^ complexes, whereas the other indicates organic radicals.

The first signal is characteristic of a Mn^2+^ (M_I_ = 5/2) -related sextet in Figure 6; the natural abundance of manganese (^55^Mn isotope) was known to be 100%. The upper inset figure shows the expansion of the 300 to 375 mT region. The arrow indicates the magnetic field at approximately 336 mT. The signal at the arrow has a narrower line width than the other signals. This indicates that the signal is different in different species. Thus, the EPR spectra composed from this sextet were attributed to Mn^2+^ peaks. The hyperfine structure of Mn^2+^ is at 6.8 to 10.7 mT (Figure 6B). The hyperfine values of the sextet are also similar to previously reported values [15]. The apparent changes in hyperfine couplings from low to high fields were found to be larger in high fields, owing to the correlation time and overlap with other broad features. A similar EPR spectrum was previously reported in apple seeds [15]. Thus, the observed signatures from these two different radical species were assigned as indicators of stable organic radicals and Mn^2+^ complexes. The asterisk features (~260 mT) are not clear at this time (Figure 6B). The EPR spectrum of Figure 6B also overlaps with the organic radical and broad features. The Mn^2+^ signals were clearly detected. The broad signal at ~260 mT is not known at this point. It was noted that the EPR intensities of dragon fruits became larger when the samples were submerged in H_2_O and dried out in a refrigerator overnight. The amplitude of the signal may be related to the oxidation and/or antioxidation of the compounds in each sample.

We further examined the EPR signal for the red peel samples. We cut out approximately 1 × 2 × 3 mm^3^ red peel (~0.0009 g) and submerged it in distilled water (~40 μL) overnight in a 5 °C refrigerator. We took ~10 μL of the solution using a glass capillary (i.d., 0.9 mm; o.d., 1.2 mm) and measured the solution. We observed the EPR signal from the distilled water extracted solution. The EPR spectrum is presented in Figure 7. The ΔH_pp_ of the EPR peak is ~1.8 mT. The wider signal indicates various radicals overlapping in the region. In addition, the results suggest that the motional states of various intermediates (radicals) differ between the dry and aqueous sample solutions. Note that the dilution is the major concern for the detection, although the amount of compounds depends on the plant sample.

Figure 8 shows the EPR spectra of various dragon fruits (red and white). The g-value of the EPR signal is approximately 2.004. The EPR spectra of the red dragon fruit peel and pulp are similar; the line widths are in the range 0.59–0.66 mT, as indicated in Table 3. However, the peel of white dragon fruits has a broader line width (ΔHpp = 1.45 mT) (Figure 8B), about twice as broad as that of the red peel. In addition, the EPR intensities of the white peel and pulp are very weak, even taking into account the weights measured. The broad EPR line may be due to the unresolved hyperfine coupling of one radical due to an inhomogeneous environment and/or the overlap of spectra from different organic radicals with different g-values and hyperfine coupling constants. Thus, one can detect the broader line in the sample [34]. Relatively sharp line widths were obtained for red and white dragon fruit seeds (Figure 8E,F). The baselines of the EPR spectra in Figure 8 are not flat due to the overlap with other features, as the organic radicals appear between the third and the fourth Mn^2+^ hyperfine couplings.

In general, the EPR line width can be influenced by two factors. One is the motion of the radical, and the other is the number of different radicals overlapping. The colored samples (peel and pulp) show very similar line widths, but not the white peel. The border line width of the white peel could have been caused by the overlapping of multiple EPR lines. These multiple signals could be due to different species in the white peel.

Figure 9 shows a bar plot of the relative area of the EPR signal for various dragon fruit parts. The peel and seeds exhibit large signal areas. The analysis of the EPR signal was obtained by taking a double integral of the spectrum. Then, each area was divided by the weight of the samples measured, because the signal intensity is proportional to the amount of the sample measured. The relative area of the seeds is approximately eight to nine times larger than that of the peel. The results suggest that the total radical concentration here is much higher than in other parts of the dragon fruits. Although the peak intensity of the white peel is very weak, the signal area of the peel is close to that of the red peel due to the broad line width. It is also interesting to note that the pulp parts show very weak signals for both.

The EPR signal area is proportional to the total concentration of radicals in the samples. The seeds have more radicals than the other samples. The seeds have larger amounts of antioxidants, as determined by HPLC. These results are consistent with those of the HPLC. In addition, the seeds are different in color and contain more phenolics than the other parts of the fruit. The results from the EPR and HPLC are consistent, although nondestructive EPR can detect biochemical intermediates in fruits. Stable intermediates, which delocalize unpaired electrons in antioxidants, are present in the seeds.

## 3. Materials and Methods

### 3.1. Samples Preparation

Two kinds of dragon fruits, red and white pulp, were purchased from the royal project shop, Chiang Mai, Thailand, in June 2020. Samples were cut out from the dragon fruits for EPR measurements and stored in a 5 °C refrigerator to dry them. Each sample was inserted into an EPR tube (o.d., 5.0 mm; i.d., 4.0 mm) for measurements. For other experiments, fresh red pulp (RP), red pulp seed (RP-S), red peel (RPe), fresh white pulp (WP), white peel (WPe), and white pulp seed (WP-S) were freeze-dried (0.07 mbar, −45 °C for 48 h) and cut into small pieces. Then, all the samples were placed in a hydroalcoholic solution at pH 2 to obtain the extracts. The obtained solutions were filtrated, and the solvent was evaporated under reduced pressure and vacuum-dried [35].

### 3.2. Total Phenolic Measurement

The total phenolic contents (TPC) of each dragon fruit extract were measured using a slightly modified Folin–Ciocalteu colorimetric assay. All the results are expressed as mg gallic acid equivalent (GAE) per g of dried weight [36].

### 3.3. Total Flavonoid Measurement

The total flavonoid contents (TFC) of each dragon fruit extract were analyzed using a modified colorimetric assay, as described by Phromnoi et al. [37]. Briefly, 150 μL of 5% sodium nitrite was mixed with 2 mL of distilled water. Then, 500 μL of dragon fruit extracts or positive control (quercetin) were added to a mixture solution. Sample mixtures were incubated at room temperature for 5 min and protected from light. Then, 10% aluminum chloride hexahydrate solution was added and the mixture was incubated for 5 min. Finally, 1 mL of 1 M sodium hydroxide was added and the total volume was adjusted to 5 mL using deionized water. The mixture solutions were then incubated for 10 min at room temperature. After incubation, the absorbance was measured at 510 nm and the TFC was expressed as the mg quercetin equivalent (QE) per g of dried weight.

### 3.4. Total Anthocyanin Measurement

The total anthocyanin content (TAC) was measured using the slightly modified pH differential method of Pengkumsri et al. [38]. Briefly, 4.0 mL of the buffer solution of pH 1.0 or 4.5 and 1 mL of samples or positive control (cyanidin chloride) were mixed and incubated at room temperature for 20 min with protection from light. After incubation, the absorbance was measured at 510 and 700 nm. TAC was expressed as the mg cyanidin chloride equivalent per g of dried weight.

### 3.5. Chromatographic Analysis of Anthocyanins

All the samples were analyzed for anthocyanins by reverse-phase HPLC with slight modifications from Pengkumsri et al. [38] using an Agilent 1200 equipped with a multiwavelength detector (Agilent Technologies Inc., Santa Clara, CA, USA). Symmetry RP18 Column (4.6 mm of diameter × 250 mm of length, 5 µm particle diameter, Waters Co., Ltd., Milford, MA, USA) was used to separate each form of anthocyanin and the detection wavelength was set at 520 nm. Phosphoric acid (3%) in acetonitrile and phosphoric acid (3%) in deionized water was used as the mobile phase at a flow rate of 1.0 mL/min. The linear gradient elution was operated from 0 to 40 min, with acetonitrile ranging from 10% to 20%. All the samples were tested in triplicate.

### 3.6. Chromatographic Analysis of Catechin and Related Compounds

Catechin and related compounds, including epicatechin, epicatechin gallate, epigallocatechin, epigallocatechin gallate, gallocatechin, gallocatechin gallate, gallic acid, and caffeine, were analyzed by reverse-phase HPLC according to the conditions of Saenjum et al. [39] using an Agilent 1200 equipped with a multiwavelength detector. Briefly, the detection wavelength was set at 210, 278, and 325 nm. The Symmetry Shield RP18 Column (4.6 mm of diameter × 250 mm of length, 5 µm particle diameter, Waters Co., Ltd.) was used to separate each form of catechin and its related compounds. The mobile phase consisted of 10% acetonitrile in 0.1% acetic acid and deionized water at a flow rate of 1.0 mL/min.

### 3.7. Determination on Inhibition Effect on Intracellular Reactive Oxygen Species (ROS) Production

The inhibitory effect of dragon fruit extracts on intracellular ROS production was determined following the modified method of Banjerdpongchai et al. [40]. Briefly, peripheral blood mononuclear cells were pretreated with 10–100 mg × L^−1^ of tested samples for 12 h. Then, 5 mM of hydrogen peroxide (H_2_O_2_) was added for 30 min to initiate ROS production. DCFH-DA solution was added to each mixture and incubated at 37 °C and 5% CO_2_ for 30 min. Fluorescence intensity was measured using a fluorescent microplate reader at an excitation wavelength of 480 nm and an emission wavelength of 525 nm. *N*-Acetyl cysteine, cyanidin 3-glucoside, and L-ascorbic acid were used as positive controls.

### 3.8. Determination of the Inhibitory Effect on Intracellular Reactive Nitrogen Species (RNS) Production and Anti-Inflammatory Activities

The inhibition of nitric oxide production was assayed using the improved method of Sirithunyalug et al. [41] and Phromnoi et al. [19], with some modifications. Initially, RAW 264.7 cells were pretreated with various concentrations of the tested samples at concentrations of 10–100 mg × L^−1^ and the positive controls, curcumin and cyanidin 3-glucoside, at concentrations of 2.5–50 mg × L^−1^. After a 12 h incubation period, LPS and IFN-γ were added to the media. After 48 h of incubation, the culture medium supernatants were collected to analyze the level of nitric oxide. Griess reagent was used to quantify the nitrile level as an indicator of nitric oxide production; the absorbance was measured at 540 nm. Moreover, cells were lysed to yield cell lysates using the CelLytic^TM^ M Cell Lysis Buffer (Sigma, C2978, Merck KGaA, Darmstadt, Germany) to perform the assay for iNOS and COX-2 using the mouse ELISA kit following the manufacturer’s protocol. The quantity of DNA was measured by the Quant-iT PicoGreen Assay (Invitrogen, P11496, Thermo Fisher Scientific Inc., MA, USA), while that of the protein produced by the HT-29 cells was analyzed using Bradford reagent (Sigma Chemical Co., Ltd., St. Louis, MO, USA).

### 3.9. EPR Measurements

A modified JEOL RE-3X 9 GHz EPR spectrometer (JEOL Ltd., Tokyo, Japan) was used for various measurements. The system was operated in X-band mode at approximately 9.43 GHz using a 100 kHz modulation frequency. All the EPR spectra were obtained from a single scan. The typical EPR settings were as follows: microwave power, 5 mW; time constant, 0.1 s; sweep time, 4 min; magnetic field modulation, 0.32 mT; and sweep width, 10 mT. The details were described in the previous report [42].

### 3.10. Statistical Analysis

IBM SPSS Statistics Baes 22.0 was used for statistical analysis. Data are expressed as mean ± standard deviation (SD). Statistical analysis was determined using one-way analysis of variance. Significant difference at the levels of *p* < 0.05 was measured in this study.

## 4. Conclusions

Anthocyanins namely, cyanidin 3-glucoside, delphinidin 3-glucoside, and pelargonidin 3-glucoside were detected in the dragon fruit peel and red pulp. Cyanidin 3-glucoside was analyzed as a major component in the dragon fruit peel and red pulp. Catechin, epicatechin, epicatechin gallate, epigallocatechin, caffeine, and gallic acid were found in the dragon fruit seed. Moreover, cyanidin 3-glucoside enriched extracts prepared from dragon fruit were found to inhibit the production of reactive oxygen species (ROS), reactive nitrogen species (RNS), inducible nitric oxide synthase (iNOS), and cyclooxygenase-2 (COX-2) without exerting cytotoxicity in this cell-based study. Additionally, non-destructive EPR detected two different paramagnetic species in dragon fruits. The line broadening of the radical in the white peel could be due to the different radical moieties present. The red peel of dragon fruits contained higher amounts of antioxidants than the white pulp of dragon fruits. The results regarding antioxidative phytochemicals and related intermediates were consistent between the EPR and HPLC measurements. Thus, HPLC and EPR spectroscopy provided useful information regarding antioxidants and related phytochemicals in dragon fruits. EPR is a useful method that can be applied for the evaluation of stable paramagnetic species and related antioxidative intermediates in bioresources or waste biomass samples. In summary, the obtained results demonstrated the potential of cyanidin 3-glucoside-enriched extract prepared from a waste biomass, dragon fruit peel, to be used as a natural active pharmaceutical ingredient (natural API) for nutraceutical and nutricosmetic products.

## Figures and Tables

**Figure 1 molecules-26-03565-f001:**
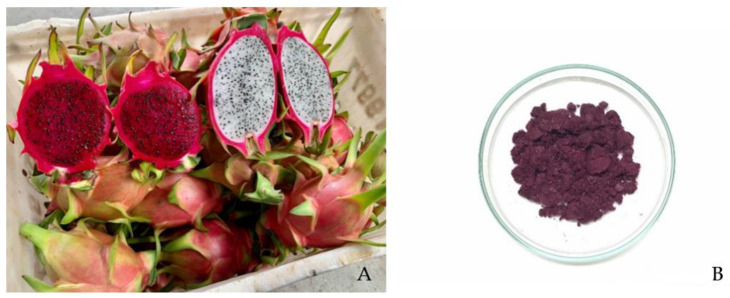
Red and white dragon fruits used in this study (**A**) and anthocyanin extract prepared from the red peel (**B**).

**Figure 2 molecules-26-03565-f002:**
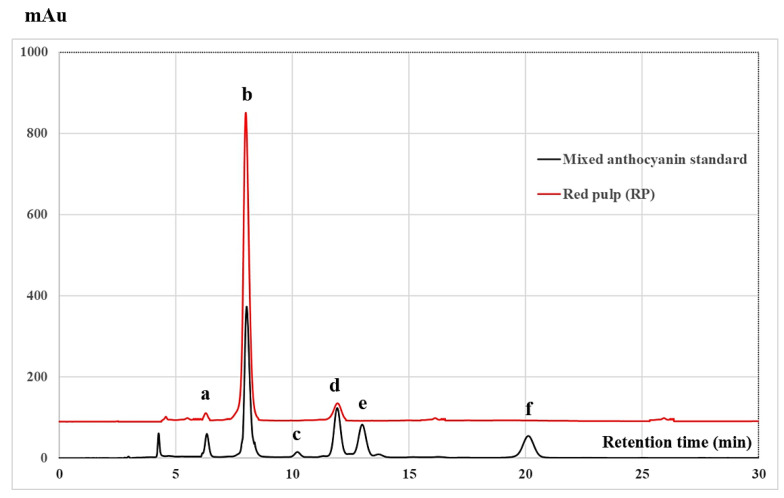
High-performance liquid chromatography (HPLC) chromatograms of mixed anthocyanin standards and red pulp (RP) extract. The peaks indicate (**a**) delphinidin 3-glucoside, (**b**) cyanidin 3-glucoside, (**c**) peonidin 3-glucoside, (**d**) pelargonidin 3-glucoside, (**e**) delphinidin, and (**f**) cyanidin.

**Figure 3 molecules-26-03565-f003:**
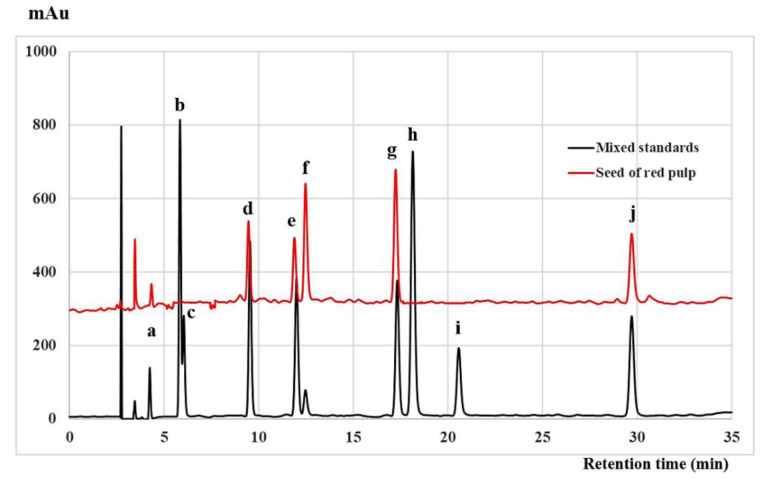
HPLC chromatograms of mixed catechin and related compounds and the extract of the seeds of red pulp (RP-S). The peaks indicate (**a**) gallic acid, (**b**) pyrogallol, (**c**) gallocatechin, (**d**) epigallocatechin, (**e**) caffeine, (**f**) catechin, (**g**) epicatechin, (**h**) epigallocatechin gallate, (**i**) gallocatechin gallate, and (**j**) epicatechin gallate.

**Figure 4 molecules-26-03565-f004:**
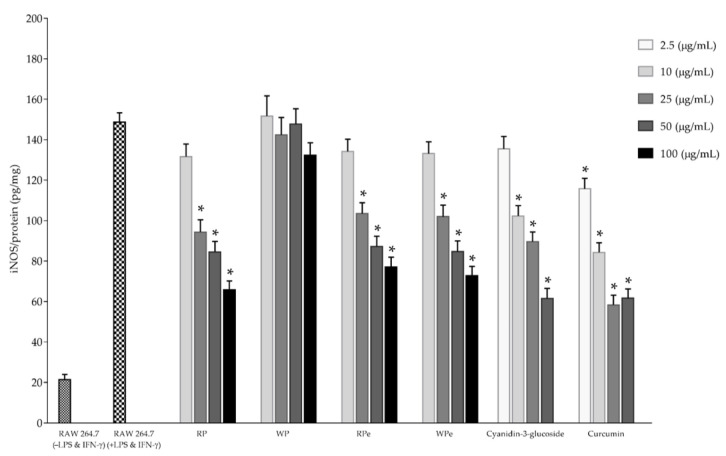
The inhibition effect on iNOS production in LPS/IFN-γ-induced RAW 264.7 cells of selected anthocyanin extracts (RP, WP, RPe, and WPe). * Significant differences at the 95% confidence interval compared to control RAW 264.7 cells treated with LPS and IFN-γ.

**Figure 5 molecules-26-03565-f005:**
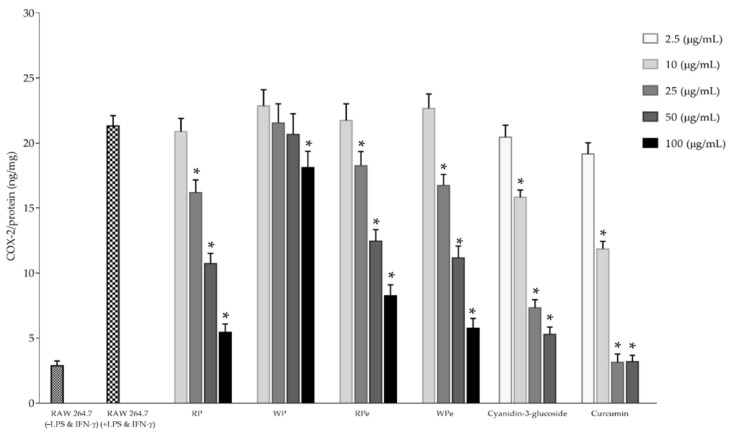
The inhibition effect on COX-2 production in LPS/IFN-γ-induced RAW 264.7 cells of selected anthocyanin extracts (RP, WP, RPe, and WPe). * Significant differences at the 95% confidence interval compared to control RAW 264.7 cells treated with LPS and IFN-γ.

**Figure 6 molecules-26-03565-f006:**
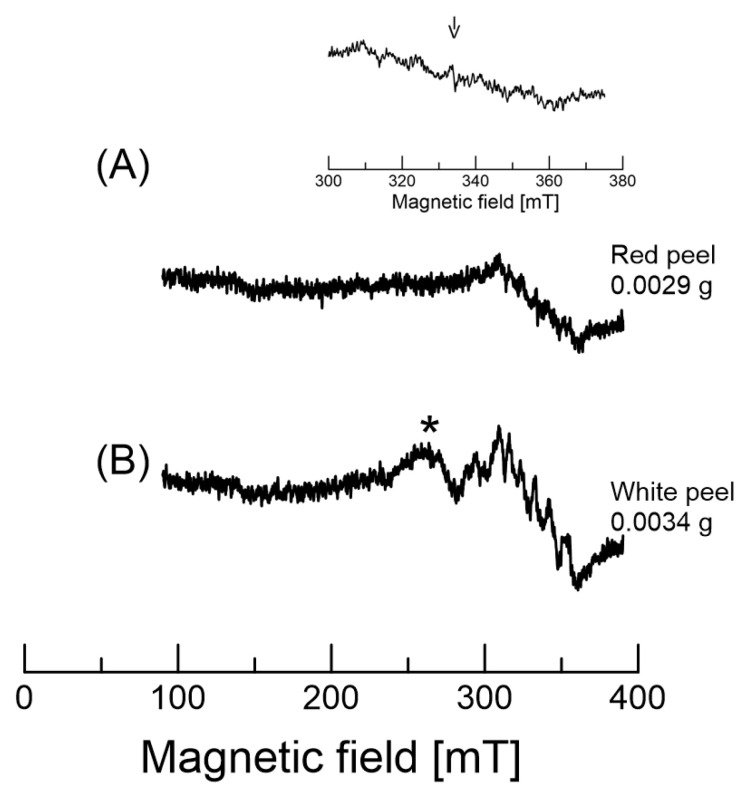
EPR spectra obtained by a wide-range scan (300 mT) of the red and white peel of dragon fruits. (**A**) shows the EPR spectrum of the red peel. The upper figure shows the expansion of the 300 to 375 mT region. The arrow indicates the magnetic field at ~336 mT. (**B**) shows the EPR spectrum of the white peel. The arrow (~335 mT) indicates organic radicals. The asterisk is not known at this point.

**Figure 7 molecules-26-03565-f007:**
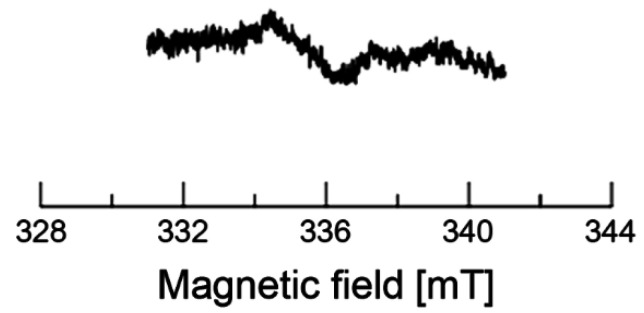
EPR spectra of red peel-related samples. Upper spectrum is for the dry red peel sample. The bottom one is for the solution of red peel samples submerged in distilled water. The bottom spectrum was accumulated four times.

**Figure 8 molecules-26-03565-f008:**
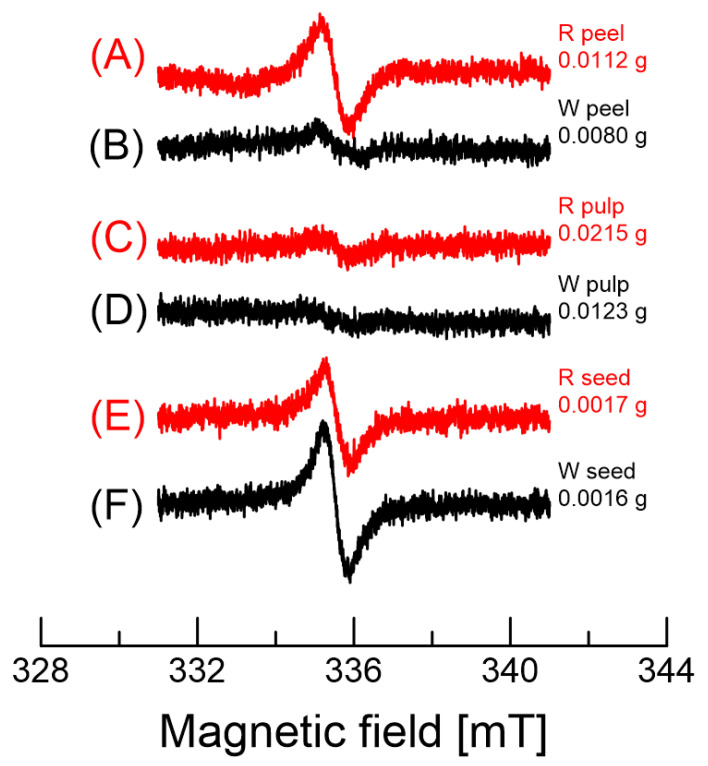
EPR spectra of various parts of dragon fruits. (**A**,**B**) show the spectra of the peel of red and white dragon fruits, respectively. (**C**,**D**) show the spectra of the pulp of red and white dragon fruits, respectively. (**E**,**F**) show the spectra of the seeds of red and white dragon fruits, respectively. The sample weight measured is indicated at the right side of each spectrum.

**Figure 9 molecules-26-03565-f009:**
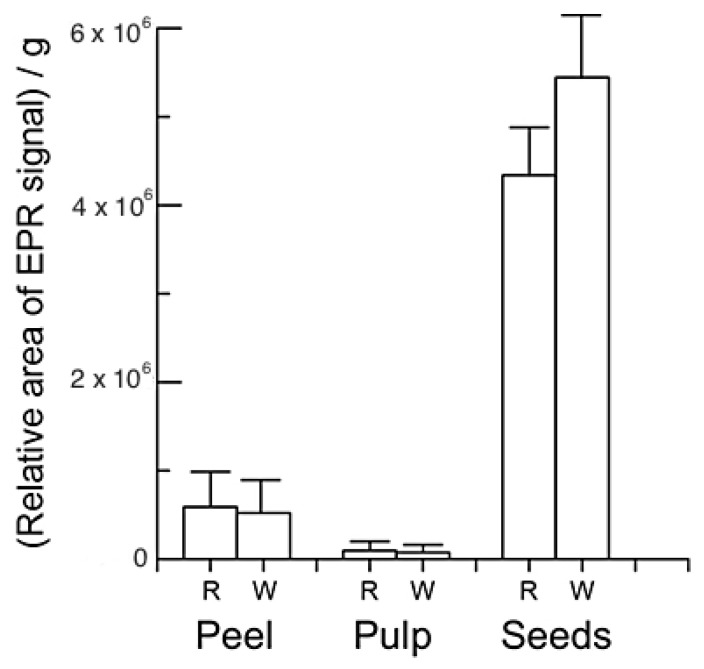
Bar plot showing the relative areas of EPR signals as a function of the part of the dragon fruit used. The area of each EPR spectrum was obtained by the double integral. The results are expressed as the mean ± standard error. The letters (R and W) represent red and white, respectively.

**Table 1 molecules-26-03565-t001:** Total phenolic, flavonoid, and anthocyanin contents.

Sample Extracts	Total Phenolic Content (TPC)	Total Flavonoid Content (TFC)	Total Anthocyanin Content (TAC)
	mg GAE/g Dried Weight	mg QE/g Dried Weight	mg CCE/g Dried Weight
Fresh red pulp (RP)	277.6 ± 14.2 ^c^	177.4 ± 12.5 ^c^	159.7 ± 8.9 ^a^
Fresh white pulp (WP)	48.4 ± 6.8 ^e^	39.5 ± 7.3 ^e^	ND
Red peel (RPe)	294.8 ± 12.9 ^c^	193.8 ± 11.7 ^c^	135.4 ± 9.3 ^b^
White peel (WPe)	207.3 ± 8.5 ^d^	142.9 ± 12.6 ^d^	106.8 ± 7.4 ^c^
Red pulp seed (RP-S)	375.1 ± 12.6 ^a^	264.4 ± 10.8 ^a^	ND
White pulp seed (WP-S)	338.7 ± 13.3 ^b^	227.6 ± 11.6 ^b^	ND

All values are expressed as means ± standard deviation (SD; *n* = 3). Different letters in each column indicate a significant difference (*p* < 0.05). ND = not detectable.

**Table 2 molecules-26-03565-t002:** The 50% inhibitory concentrations (µg/mL) of extracts for the production of reactive nitrogen species (RNS) and reactive oxygen species (ROS).

Sample Extracts	IC_50_ (µg/mL)	
	RNS	ROS
Fresh red pulp (RP)	13.72 ± 1.08 ^b^	18.73 ± 1.17 ^d^
Fresh white pulp (WP)	>100	>100
Red peel (RPe)	17.32 ± 1.26 ^c^	24.37 ± 1.41 ^e^
White peel (WPe)	22.49 ± 1.15 ^d^	27.85 ± 1.83 ^e^
Red pulp seed (RP-S)	29.78 ± 1.42 ^e^	65.42 ± 2.53 ^f^
White pulp seed (WP-S)	32.49 ± 1.36 ^e^	73.08 ± 2.17 ^e^
Cyanidin 3-glucoside	6.25 ± 0.58 ^a^	8.48 ± 0.63 ^b^
Curcumin	5.84 ± 0.73 ^a^	ND
N-Acetylcysteine	ND	5.18 ± 0.42 ^a^
L-Ascorbic acid	ND	12.54 ± 0.79 ^c^

All values are expressed as means ± standard deviations (SD; *n* = 3). Different letters in each column indicate a significant difference (*p* < 0.05). ND = not detectable.

**Table 3 molecules-26-03565-t003:** EPR peak-to-peak line width of dragon fruits.

	Red			White		
	Peel	Pulp	Seed	Peel	Pulp	Seed
**ΔHpp (mT)**	0.66 ± 0.06	0.64 ± 0.05	0.61 ± 0.06	1.45 ± 0.13	0.59 ± 0.05	0.61 ± 0.06

All values are expressed as means ± error (*n* = 3).

## Data Availability

The original contributions generated for this study are included in the article; the data presented in this study are available on request from the corresponding author.

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
