# Peer review of "Antioxidative and Anti-Inflammatory Phytochemicals and Related Stable Paramagnetic Species in Different Parts of Dragon Fruit"

_molecules, 2021, doi:10.3390/molecules26123565_

Round 1

Reviewer 1 Report

The manuscript by C. Saenjum, T. Pattananandecha and K. Nakagawa entitled ”Antioxidative and anti-inflammatory phytochemicals and related stable paramagnetic species in different parts of dragon fruit” concerns the antioxidant and anti-inflammatory phytochemicals and paramagnetic species in dragon fruit. However, the biggest drawback of the reviewed manuscript is the lack of novelty. In the last few years a large number of studies have been published (also by the authors of this paper) in this subject area. Here are some examples: 10.1016/j.foodchem.2021.129426; 10.1088/1742-6596/1836/1/012069; 10.1016/j.fbio.2021.100888; 10.3390/molecules26082158.

I have the following comments and suggestions for the authors:

  1. Line 18: The authors claim that “Catechin, epicatechin, epicatechin gallate, epigallocatechin, caffeine, and gallic acid were found in the dragon fruit seed”, but it is mentioned in the Introduction that the catechin and epicatechin has already been discovered earlier in dragon fruit by Abdul Hamid, A. et al. Is it necessary to specify the catechin and epicatechin in the abstract?
  2. Line 80: Why pH 2 was chosen?
  3. Line 158-159: Please, specify again all the required abbreviations. It is uncomfortable to find them.
  4. Line 218: change the “cyatotoxicity” to “cytotoxicity”.

I should draw the attention to the fact that the percentage of self-citation for Kouichi Nakagawa and Chalermpong Saenjum reaches almost 20%.

Unfortunately, I didn’t find supplementary materials.

Author Response

Dear,

The Editor Molecules

Submission of revised manuscript (R1) for publication

Very much thank you for your e-mail of May 26th, 2021 concerning our manuscript (Molecules-1247596). We appreciate the useful suggestions and comments made by the reviewers. We attempted to respond carefully to the specific suggestions and English editing by English editing team (MDPI). We intend to publish an article entitled “Antioxidative and Anti-Inflammatory Phytochemicals and Related Stable Paramagnetic Species in Different Parts of Dragon Fruit” (Molecules-1247596R1) in your esteemed journal as a Letter to the Editor. On behalf of all the contributors, I would like to response to the comments as followed:

Responses to the reviewer 1:

  1. The authors claim that “Catechin, epicatechin, epicatechin gallate, epigallocatechin, caffeine, and gallic acid were found in the dragon fruit seed”, but it is mentioned in the Introduction that the catechin and epicatechin has already been discovered earlier in dragon fruit by Abdul Hamid, A. et al. Is it necessary to specify the catechin and epicatechin in the abstract?

Response 1: Thank you for your comment. According to your comment, catechin and epicatechin was deleted from the abstract.

  1. Line 80: Why pH 2 was chosen?

Response 2: Thank you for your comment. Anthocyanins are stable and have a good solubility in acidic and condition.

  1. Line 158-159: Please, specify again all the required abbreviations. It is uncomfortable to find them.

Response 3: Thank you for your comments. All of abbreviations were specified as shown in line 172-173.

  1. Line 218: change the “cyatotoxicity” to “cytotoxicity”.

Response 4: Thank you for your comment and was corrected as shown in line 271.

  1. I should draw the attention to the fact that the percentage of self-citation for Kouichi Nakagawa and Chalermpong Saenjum reaches almost 20%.

Response 5: Thank you for your comment. The percentage of  self-citation for Kouichi Nakagawa and Chalermpong Saenjum was improved to lower than 20%

  1. Unfortunately, I didn’t find supplementary materials.

Response 6: Thank you for your comment. The supplementary materials including CSV files for anthocyanins, catechin and related compounds were attached.

With Best Regards,

(Assist. Prof. Dr.Chalermpong Saenjum)

Corresponding contributor:

Department of Pharmaceutical Sciences, Faculty of Pharmacy, Chiang Mai University, Chiang Mai, 50200, Thailand

Phone numbers +66-5394-4312, +66-89-9504227

E-mail address chalermpong.saenjum@gmail.com; chalermpong.s@cmu.ac.th

Reviewer 2 Report

Executive Summary

The manuscript titled “Antioxidative and Anti-inflammatory Phytochemicals and Related Stable Paramagnetic Species in Different Parts of Dragon Fruit” investigated the healthy related effect dragon fruit may have through scientific experiments. Overall, the article is well written with good discussion and logic. The authors may perform minor revisions to further improve the quality before publication.

Major Comments

  • There are no major comments.

Minor Comments

  • Figure 4: it is almost impossible to see the while columns, which indicates 2.5 µg/mL. Please change the color or use border color to improve the reader's friendliness.
  • Figure 6 and Figure 7 should overlay to show potential differences, especially for water as a control.
  • Table 3: please use distilled water as the control group, since “the states of intermediates (radicals) are different between the dry samples and extracted aqueous solution”.

Author Response

Dear,

The Editor Molecules

Submission of revised manuscript (R1) for publication

Very much thank you for your e-mail of May 26th, 2021 concerning our manuscript (Molecules-1247596). We appreciate the useful suggestions and comments made by the reviewers. We attempted to respond carefully to the specific suggestions and English editing by English editing team (MDPI). We intend to publish an article entitled “Antioxidative and Anti-Inflammatory Phytochemicals and Related Stable Paramagnetic Species in Different Parts of Dragon Fruit” (Molecules-1247596R1) in your esteemed journal as a Letter to the Editor. On behalf of all the contributors, I would like to response to the comments as followed:

Responses to the reviewer 2:

  1. Figure 4: it is almost impossible to see the while columns, which indicates 2.5 µg/mL. Please change the color or use border color to improve the reader's friendliness.

Response 1: Thank you for your comment. The color of 2.5 µg/mL column was changed to improve the reader’s friendliness.

  1. Figure 6 and Figure 7 should overlay to show potential differences, especially for water as a control.

Response 2: Thank you for your comments.

With Best Regards,

(Assist. Prof. Dr.Chalermpong Saenjum)

Corresponding contributor:

Department of Pharmaceutical Sciences, Faculty of Pharmacy, Chiang Mai University, Chiang Mai, 50200, Thailand

Phone numbers +66-5394-4312, +66-89-9504227

E-mail address chalermpong.saenjum@gmail.com; chalermpong.s@cmu.ac.th

Reviewer 3 Report

Dear Author(s)

After an exhaustive revision, the manuscript is Reconsider after major revision (control missing in some experiments). In general, the study is closely connected to the journal's objectives. The study is very interesting. However, the authors need to make some changes and modify parts in the manuscript, mainly in the section “Results and Discussion”, since it is very poor.

In the following pages, I give a detailed revision of the manuscript.

Best regards

ABSTRACT

The abstract is good. Only, the authors need to add the main results in terms of numbers, %, among others.

  1. INTRODUCTION

General comments

The introduction is good, since it starts from general to particular. The English is good.

The introduction needs to clarify some details and update the references. Specifically, the authors should search references from the last three years, since the authors added one study from 2021, and the others correspond to studies from 2017 backwards. Update the references will allow the authors to enhance the introduction with more current data, as well as other sections of the study.

My observations:

Line 34. "It is native to the tropical forest region of Mexico a ..."

The authors should remove the word "the".

Lines 36-44. The references [3-5] are good. However, the authors should add more current references; references [3] and [4] are old.

Lines 44-47. What are the references? Are the lines referring to some study with pitaya?

Lines 54-65. What is the relationship between these lines? The authors give an introduction to EPR, but the authors not give a clear focus to studies on fruits or bioactive components, and then the authors write about "Numerous plants contain various antioxidants to ...". What is the focus of EPR on foods? What is the relationship between EPR and antioxidants?. The reviewer also does not understand the next lines about HPLC, what is the relationship between EPR and HPLC? Can EPR and HPLC be complemented?. The authors must rewrite these lines and organize them to give continuity towards the objective of the study.

Lines 66-71. The objective is clear, but the previous lines must be reformulated to clarify the use of EPR in fruits and bioactive components and how HPLC can give other complementary results.

  1. MATERIALS AND METHODS

General comments

This section is clear. The English is good. The authors need fix to add references.

The authors must add a Figure that represents all the methodology in the section Materials and Methods. This Figure will help to understand the methodology.

This section needs a point on Statistical Analysis.

My observations:

2.1. Samples preparation

Line 79. "were freeze-dried and..."

What are the conditions?

Lines 79-84. What is the reference?

2.3. Total flavonoid measurement

Line 89. "described by Phromnoi et al. (2019) [15]."

It is correct format? or "described by Phromnoi et al. [15]."?

2.4. Total anthocyanin measurement

Line 100. "method of Pengkumsri et al. (2015) [16]."

It is correct format? or "method of Pengkumsri et al. (2015) [16]."?

2.5. Chromatographic analysis of anthocyanins

Line 107. "from Pengkumsri et al. (2015) [16]"

It is correct format? or "from Pengkumsri et al. (2015) [16]"?.

2.6. Chromatographic analysis of catechin and related compounds

2.7. Determination on inhibition effect on intracellular reactive oxygen species (ROS) production

2.8. Determination of the inhibitory effect on intracellular reactive nitrogen species (RNS) production and anti-inflammatory activities

Please,

The same problem that 2.3, 2.4, and 2.5 with the format in the references.

  1. RESULTS AND DISCUSSION

General comments

"Results and Discussion" is characterized by a description of the results, comparison with other studies and explication (discussion) of the results obtained with respect to other studies.

The description of results starts is incomplete, without detailed. Moreover, there is comparison with other studies, but without much detail. There is no discussion in the subsections.

My observations:

3.1. TPC, TFC, and TAC content

The description of the results is not complete, since it is very simple. The authors need to write about the each components and the relation with each sample (what is the concentration?. Additionally, the authors must explain why the values in "pulp seed" (differences statistically) are greater than "peel" or "pulp"?. The authors need to compare the results with other studies (similar, increase or decrease), and later, the authors need to explain (discussion) the differences or similarities of results (with other studies).

3.2. Chromatographic analysis of antioxidative phytochemicals

Lines 169-177. The description of the results needs to be completed with numerical data of the anthocyanins (what is the concentration?). The comparison with other studies is good, but it would be more interesting with the values ​​found by Fan et al. (2020) [21] (also, check citation format), [19] and [10]. However, there is no discussion regarding the behavior of the results, comparison between samples and discussion of similarity or differences of results with other studies.

Line 176. "coloration of dragon fruit [19]."

What monomers were found in this study? delphinidin 3-O-beta-D-glucoside 5-O- (6-coumaroyl-beta-D-glucoside)? The authors should clarify which monomers correspond to study [21] and [19].

Lines 176-177. "Moreover, cyanidin 3-O-glucoside, cyanidin 3,5-O-glucoside, and pelargonidin 3,5-O-glucoside are also found in the skin of dragon fruit [10]".

These lines confuse to the reviewer. Were these monomers from the study [10]? Were they found in the present study? or is it just a reference that these monomers also exist in dragon fruit?

Lines 178-184. The description of the results needs to be completed with numerical data (what is the concentration?). The comparison with other studies is good, but it would be more interesting with the values ​​found by Adnan et al. (2011) [9] (also, check citation format). However, there is no discussion regarding the behavior of the results, comparison between samples and discussion of similarity or differences of results with other studies.

3.3. Determination of the inhibitory effects on intracellular ROS and RNS production

The description of the results needs to be completed with numerical data. The comparison with other studies is good, but it would be more interesting with the values found by Wang et al. (1999) [22] and Kuskoski et al. (2004) [23] (also, check citation format).

Lines 205-208. The discussion is interesting, but the authors should link these lines with the dragon fruits.

3.4. Determination of anti-inflammatory activities

The description of the results is very poor, it does not indicate in terms of data which components are better or worse, the authors must detail. The comparison with the results of previous studies is very clear, more complete. However, the authors should detail the study by Limtrakul et al. [26] (also, check citation format). The discussion is still very general, it does not indicate the reason for the results obtained, only focuses it in a general context.

Lines 241-218. These lines are repetitive.

3.5. EPR of dragon fruits

The description of results is incomplete. What happens in Figure 6A? The description of Figure 6B is good. The comparison with other studies is poor, there is not much comment on [12]. Are there more current studies?

Lines 247. “…organic radicals…”

What type of organic radicals?

Lines 255-259. The lines are repetitive.

Lines 261-262. The discussion is very poor.

Lines 271-275. Was this comparison explained in materials and methodology?

Lines 274-275. Why? The authors need to explain reasons for the behavior.

Lines 276-288. These lines deserve an explanation more related to the scientific, since it is important to mention each aspect of the behavior of the samples. A comparison of results or a discussion is not denoted.

Lines 297-305. The description is good, but it needs more detail. The discussion is poor.

  1. CONCLUSIONS

The conclusions should be improved from the changes made in the manuscript.

REFERENCES

The authors not followed the author's guide for references.

Author Response

Dear,

The Editor Molecules

Submission of revised manuscript (R1) for publication

Very much thank you for your e-mail of May 26th, 2021 concerning our manuscript (Molecules-1247596). We appreciate the useful suggestions and comments made by the reviewers. We attempted to respond carefully to the specific suggestions and English editing by English editing team (MDPI). We intend to publish an article entitled “Antioxidative and Anti-Inflammatory Phytochemicals and Related Stable Paramagnetic Species in Different Parts of Dragon Fruit” (Molecules-1247596R1) in your esteemed journal as a Letter to the Editor. On behalf of all the contributors, I would like to response to the comments as followed:

Responses to the reviewer 3:

  1. Line 34. "It is native to the tropical forest region of Mexico a ..." The authors should remove the word "the".

Response 1: Thank you for your comment. The word “the” was removed as shown in line 34.

  1. Lines 36-44. The references [3-5] are good. However, the authors should add more current references; references [3] and [4] are old.

Response 2: Thank you for your comment. The updated references number 3, 5, and 6 were added to the introduction part.

  1. Lines 44-47. What are the references? Are the lines referring to some study with pitaya?

Response 3: Thank you for your comment. The references number 7 and 8 were added as shown in line 47.

  1. What is the relationship between these lines? The authors give an introduction to EPR, but the authors not give a clear focus to studies on fruits or bioactive components, and then the authors write about "Numerous plants contain various antioxidants to ...". What is the focus of EPR on foods? What is the relationship between EPR and antioxidants?. The reviewer also does not understand the next lines about HPLC, what is the relationship between EPR and HPLC? Can EPR and HPLC be complemented?. The authors must rewrite these lines and organize them to give continuity towards the objective of the study.

Response 4: Thank you for your comment. Line 62-68 was rewrite to “The antioxidative processes result in the production of antioxidant intermediates. Various antioxidants contain OH groups and conjugated double bonds. The delocalization of unpaired spin forms stable intermediates which are detected by EPR. Additionally, high-performance liquid chromatography (HPLC) is used to separate, identify, and quantitate various antioxidants in plants. However, there are no reports describing the distribution of antioxidative phytochemicals and related intermediates in dragon fruits”.

  1. Lines 66-71. The objective is clear, but the previous lines must be reformulated to clarify the use of EPR in fruits and bioactive components and how HPLC can give other complementary results.

Response 5: Thank you for your comment. The first sentence was reformulated to “the paramagnetic species in each part of the dragon fruit were examined using EPR and antioxidative phytochemicals were analyzed HPLC” (as shown in line 69-70).

  1. Line 79. "were freeze-dried and..." What are the conditions?

Response 6: Thank you for your comment. The condition of freeze-dried was 0.07 mbar,    -45 °C for 48 h (as shown in line 82).

  1. Lines 79-84. What is the reference?

Response 7: Thank you for your comment. The reference number 16 was added (as shown in line 85)

  1. Line 89. "described by Phromnoi et al. (2019) [15]." It is correct format? or "described by Phromnoi et al. [15]."?

Response 8: Thank you for your comment. The reference was edited to “Phromnoi et al. [18]” (as shown in line 92).

  1. Line 100. "method of Pengkumsri et al. (2015) [16]." It is correct format? or "method of Pengkumsri et al. (2015) [16]."?

Response 9: Thank you for your comment. The reference was edited to “Pengkumsri et al.  [19]” (as shown in line 103).

  1. Line 107. "from Pengkumsri et al. (2015) [16]." It is correct format? or "method of Pengkumsri et al. (2015) [16]."?

Response 10: Thank you for your comment. The reference was edited to “Pengkumsri et al.  [19]” (as shown in line 110).

  1. Chromatographic analysis of catechin and related compounds.

Response 11: Thank you for your comment. The reference was edited to “Saenjum et al.  [20]” (as shown in line 120).

  1. Determination on inhibition effect on intracellular reactive oxygen species (ROS) production.

Response 12: Thank you for your comment. The reference was edited to “Banjerdpongchai et al. [21]” (as shown in line 128).

  1. Determination of the inhibitory effect on intracellular reactive nitrogen species (RNS) production and anti-inflammatory activities.

Response 13: Thank you for your comment. The reference was edited to “Sirithunyalug et al. [22] and Phromnoi et al. [19]” (as shown in line 138).

  1. TPC, TFC, and TAC content. The description of the results is not complete, since it is very simple. The authors need to write about the each components and the relation with each sample (what is the concentration?. Additionally, the authors must explain why the values in "pulp seed" (differences statistically) are greater than "peel" or "pulp"?. The authors need to compare the results with other studies (similar, increase or decrease), and later, the authors need to explain (discussion) the differences or similarities of results (with other studies).

Response 14: Thank you for your comment. More additional discussion and references were added in line 166-168 [25], 170-177 [26-28], and 178-183 [29].

  1. Lines 169-177. The description of the results needs to be completed with numerical data of the anthocyanins (what is the concentration?). The comparison with other studies is good, but it would be more interesting with the values found by Fan et al. (2020) [21] (also, check citation format), [19] and [10]. However, there is no discussion regarding the behavior of the results, comparison between samples and discussion of similarity or differences of results with other studies.

Response 15: Thank you for your comment. More additional results, discussion, and references were added in line 206-207, 217-218 [13], and 218-220 [29, 31].

  1. Line 176. "coloration of dragon fruit [19]."

What monomers were found in this study? delphinidin 3-O-beta-D-glucoside 5-O- (6-coumaroyl-beta-D-glucoside)? The authors should clarify which monomers correspond to study [21] and [19].

Response 16: Thank you for your comment. Only Fan et al. [30] was cited and reference number 19 was removed.

  1. Lines 176-177. "Moreover, cyanidin 3-O-glucoside, cyanidin 3,5-O-glucoside, and pelargonidin 3,5-O-glucoside are also found in the skin of dragon fruit [10]".

These lines confuse to the reviewer. Were these monomers from the study [10]? Were they found in the present study? or is it just a reference that these monomers also exist in dragon fruit?

Response 17: Thank you for your comment. Line 176-177 was removed.

  1. Lines 178-184. The description of the results needs to be completed with numerical data (what is the concentration?). The comparison with other studies is good, but it would be more interesting with the values found by Adnan et al. (2011) [9] (also, check citation format). However, there is no discussion regarding the behavior of the results, comparison between samples and discussion of similarity or differences of results with other studies.

Response 18: Thank you for your comment. The additional discussion and references were added in line 223-236.

  1. Determination of the inhibitory effects on intracellular ROS and RNS production

The description of the results needs to be completed with numerical data. The comparison with other studies is good, but it would be more interesting with the values found by Wang et al. (1999) [22] and Kuskoski et al. (2004) [23] (also, check citation format).

Lines 205-208. The discussion is interesting, but the authors should link these lines with the dragon fruits

Response 19: Thank you for your comment. The additional discussion and references were added in line 247-250, 262-264, and 268-269.

  1. Determination of anti-inflammatory activities

The description of the results is very poor, it does not indicate in terms of data which components are better or worse, the authors must detail. The comparison with the results of previous studies is very clear, more complete. However, the authors should detail the study by Limtrakul et al. [26] (also, check citation format). The discussion is still very general, it does not indicate the reason for the results obtained, only focuses it in a general context.

Lines 241-218. These lines are repetitive.

Response 20: Thank you for your comment. The additional discussion and references were added in line 290-293 and 295-300.

  1. EPR of dragon fruits

The description of results is incomplete. What happens in Figure 6A? The description of Figure 6B is good. The comparison with other studies is poor, there is not much comment on [12]. Are there more current studies?

Response 21: Thank you for your comment. The additional discussion was added in line 320-323.

  1. Lines 271-275. Was this comparison explained in materials and methodology?

Response 22: Thank you for your comment. The additional comparison explained in materials and methodology was added in line 347-350.

  1. Lines 274-275. Why? The authors need to explain reasons for the behavior.

Response 23: Thank you for your comment. The wider signal indicates various radicals overlapping in the region. In addition, the results suggest that the motional states of various intermediates (radicals) differ between the dry and aqueous sample solutions. Note that the dilution is the major concern for the detection, although the amount of compounds depends on the plant sample (as shown in line 352-356).

  1. Lines 276-288. These lines deserve an explanation more related to the scientific, since it is important to mention each aspect of the behavior of the samples. A comparison of results or a discussion is not denoted.

Response 24: Thank you for your comment. The additional discussion was added in lines 378-382.

  1. Lines 297-305. The description is good, but it needs more detail. The discussion is poor.

Response 25: Thank you for your comment. The additional discussion was added in lines 393-399.

With Best Regards,

(Assist. Prof. Dr.Chalermpong Saenjum)

Corresponding contributor:

Department of Pharmaceutical Sciences, Faculty of Pharmacy, Chiang Mai University, Chiang Mai, 50200, Thailand

Phone numbers +66-5394-4312, +66-89-9504227

E-mail address chalermpong.saenjum@gmail.com; chalermpong.s@cmu.ac.th

Round 2

Reviewer 1 Report

All necessary changes have been revised to the manuscript.

Reviewer 3 Report

Dear Author(s)

After an exhaustive revision, the manuscript is Accept in present form. The resubmitted manuscript has been completely improved compared to its previous version. Therefore, the manuscript can be published in “Molecules”.

Best regards.